

# Prognostic significance and pathogenesis of RFC3 gene expression in diffuse large B-cell lymphoma

Zuguo Tian, Shuiyu Liu and Chunlan Weng

Department of Hematology, Affiliated Hospital of Zunyi Medical University, Zunyi, Guizhou, China

## ABSTRACT

**Background**. Diffuse large B-cell lymphoma (DLBCL) is the most common non-Hodgkin lymphoma (NHL) subtype in adults. Dysregulation of replication factor C subunit 3 (RFC3) gene expression were associated with disease progression and poor prognosis in various cancer types. However, its significance in DLBCL remains largely unexplored. This study aimed to characterize RFC3 expression patterns, clinical relevance, functional mechanisms, and potential therapeutic implications in DLBCL.

**Methods**. Multi-omics analyses were performed using data extracted from the Gene Expression Omnibus (GEO) project (GSE181063, GSE10846, GSE32918, GSE31312, GSE32018, and GSE12453) and The Cancer Genome Atlas (TCGA). RFC3 expression was validated *via* immunohistochemistry (IHC) in DLBCL samples. Survival analysis was conducted using the Kaplan–Meier method. The chi-square test was used to assess the association between RFC3 expression and clinical characteristics of DLBCL. Gene Set Enrichment Analysis (GSEA) was employed to identify tumor signaling pathways associated with RFC3. Immune infiltration was evaluated using the Immuno-Oncology Biological Research (IOBR) package. Drug sensitivity analysis was performed using the oncoPredict package, and immunotherapy response was assessed *via* the IMvigor210 dataset. Pan-cancer analysis was conducted using the easyTCGA and TCGAplot packages available on the R software.

**Results**. RFC3 expression was significantly upregulated in DLBCL. High RFC3 expression was closely associated with poor prognosis, adverse clinical features, and adverse tumor microenvironment characteristics in DLBCL patients. Furthermore, multiple tumor proliferation and cancer-related signaling pathways were significantly enriched in the high RFC3 expression group. The pan-cancer analysis also revealed elevated RFC3 expression across several tumor types. Elevated RFC3 expression was strongly correlated with worse tumor prognosis.

**Conclusions**. RFC3 may serve as a novel prognostic biomarker and a potential therapeutic target for DLBCL. Further investigations into the mechanisms underlying RFC3 dysregulation may provide important insights for future diagnostic and therapeutic strategies.

Corresponding author
Zuguo Tian, thomeland@yeah.net

## INTRODUCTION

Diffuse large B-cell lymphoma (DLBCL) is the most common subtype of non-Hodgkin lymphoma (NHL) in adults (*Swerdlow et al., 2016*). Given the biological and clinical heterogeneity of diffuse large B-cell lymphoma (DLBCL), treatment responses and survival outcomes vary significantly among patients, even within the same International Prognostic Index (IPI) risk category (*Ruppert et al., 2020*). The introduction of the rituximab, cyclophosphamide, doxorubicin, vincristine, and prednisone (R-CHOP) chemotherapy regimen used to treat DLBCL has improved survival in DLBCL patients (*Poeschel et al., 2019*). However, some DLBCL patients still experience refractory or relapsed disease, resulting in poor prognosis (*Camicia, Winkler & Hassa, 2015*). Therefore, there is a need to identify novel biomarkers to predict treatment response in DLBCL to optimize treatment strategies according to the patient's needs.

Replication factor C (RFC) is a conserved multi-protein complex that loads the sliding clamp proliferating cell nuclear antigen (PCNA) onto DNA, essential for replication and repair (*Cullmann et al., 1995*), its subunits are encoded by RFC genes. These genes are widely conserved across eukaryotes (*Tomida et al., 2008*). Studies have shown that certain RFC subunits are involved in promoting tumor cell proliferation across various cancers. For example, Replication factor C subunit 5 (RFC5) is upregulated in colorectal cancer (CRC), which is associated with a poor prognosis. It promotes tumor growth through the circ_0038985/miR-3614-5p axis and the VEGFa/VEGFR2/ERK pathway (*Yao et al., 2023*). Replication factor C subunit 4 (RFC4) facilitates the progression of nasopharyngeal carcinoma (NPC) through the modulation of HOXA10, which in turn promotes cellular proliferation and tumor growth (*Guan et al., 2023*). RFC2 is overexpressed in hepatocellular carcinoma (HCC) and is associated with advanced stages of the disease and poor survival outcomes. It promotes cell proliferation and migration through mechanisms involving the cell cycle and DNA replication, making it a potential prognostic biomarker (*Ji, Li & Wang, 2021*). Replication factor C subunit 3 (RFC3), another subunit within the RFC complex, plays a crucial role in DNA replication and repair processes and has been associated with various types of cancer. RFC3 is overexpressed in colorectal cancer (CRC), promotes tumor growth by binding to KIF14. This interaction enhances cellular proliferation, migration, and angiogenesis while inhibiting apoptosis (*Yu et al., 2024*). A recent study examined its function in cervical cancer, revealing that the silencing of RFC3 inhibited cell growth, migration, and invasion. These findings suggest potential therapeutic implications for targeting RFC3 in the treatment of cervical cancer (*Koh & Park, 2025*). However, the role of RFC3 in DLBCL remains unclear.

Therefore, this study aimed to analyze RFC3 expression in DLBCL to assess its prognostic significance and its role in DLBCL pathogenesis.

## MATERIALS AND METHODS

### Data collection

Gene expression profiles and clinical characteristics of DLBCL patients were extracted from the Gene Expression Omnibus (GEO) (https://www.ncbi.nlm.nih.gov/geo/) database. The

**Table 1  Data included in present analysis.**

| Dataset | Platform | Cases included | Details |
|---|---|---|---|
| GSE181063 | GPL14951 | 732 | DLBCL |
| GSE10846 | GPL570 | 414 | DLBCL |
| GSE32918 | GPL14951 | 167 | DLBCL |
| GSE31312 | GPL570 | 470 | DLBCL |
| GSE32018 | GPL6480 | 29 | 22 DLBCL/7 Lymph-node |
| GSE12453 | GPL570 | 21 | 11 DLBCL/10 B-cells |
| Samples collected | / | 23 | 15DLBCL/8 Lymph-node |

RFC3 expression and survival were extracted from the GSE181063 (*Painter et al., 2019*), GSE10846 (*Cardesa-Salzmann et al., 2011*), GSE32918 (*Care et al., 2014*), and GSE31312 (*Xu-Monette et al., 2020*) datasets, while the GSE32018 (*Gomez-Abad et al., 2011*) and GSE12453 (*Weniger et al., 2018*) were utilized for the differential expression analysis of RFC3 in DLBCL. All data were preprocessed and analyzed using the R software (version 4.2) as shown in Table 1. Only the data of patients treated with the R-CHOP chemotherapy regimen who had all relevant clinical characteristics reported in the database were included in this study. RFC3 expression Microarray data extracted from the GEO database (GSE32018, GSE12453) and the Gene Expression Profile Interaction Analysis database (GEPIA, http://gepia.cancer-pku.cn) were used to analyze the differential expression of RFC3 between DLBCL and control samples. The Cancer Genome Atlas (TCGA) and Genotype-Tissue Expression (GTEx) projects were used to facilitate the analysis of RFC3 expression in various tumors.

## Immunohistochemistry

Fifteen tissue samples of DLBCL lymph nodes and eight samples of normal lymph node tissue were obtained. Immunohistochemistry (IHC) was used to examine the RFC3 expression levels within the tissues as follows. All tissue samples were sectioned into two $\mu$m slices and deparaffinized using xylene and ethanol. Antigen retrieval was performed at high temperature in an ethylenediaminetetra-acetic acid (EDTA) solution (pH = 9), followed by cooling to room temperature (25 °C). Endogenous peroxidase activity was blocked using a 3% hydrogen peroxide solution. Sections were then incubated overnight at 4 °C with the primary antibody against RFC3 (1:300; Biodragon). After primary incubation, a secondary antibody was applied for 8 min. The colorimetric signal was developed using diaminobenzidine (DAB), and nuclei were counterstained with hematoxylin for 1 min. The slides were subsequently dehydrated, cleared, and mounted. ImageJ software was used to analyze integrated optical density (IOD) values. Statistical analysis and data visualization were performed using GraphPad Prism 9.0 to compare RFC3 expression between DLBCL and normal lymph node tissues.

## Ethical considerations

This study was approved by Ethics Committee of Affiliated Hospital of Zunyi Medical University (Ethics No:KLL-2024-020) (Zunyi, China) (Supplementary Material 1). The research utilized anonymized samples collected from patients previously diagnosed and treated, with no direct contact or intervention involved. Given the retrospective nature and the inability to obtain informed consent from all individuals, the committee granted a waiver of consent. All procedures complied with ethical standards to ensure patient confidentiality and data protection.

## Prognostic significance of RFC3 expression in DLBCL

The pROC R package (*Robin et al., 2011*) was used to calculate the optimal cut-off value for RFC3 expression to stratify DLBCL patients into high and low RFC3 expression groups. Kaplan–Meier survival analysis was then conducted to assess differences in overall survival (OS) between the two groups. To ensure the robustness of the findings, validation analyses were performed using multiple independent datasets, including GSE10846, GSE32918 and GSE31312. Survival analysis and visualization were carried out using the R packages survival (*Therneau, 2023*) and survminer (*Kassambara, Kosinski & Biecek, 2021*).

## Analysis of the correlation between RFC3 expression and clinical characteristics

The Chi-square test was used to analyze the correlation between RFC3 expression and clinical characteristics of DLBCL patients including age, gender, cell of origin (COO), lactate dehydrogen-ase levels (LDH), Ann Arbor stage, Eastern Cooperative Oncology Group (ECOG) Performance Status, B symptoms (Bs), extronodal sites and IPI. The Crasstable function of the gmodels R package (*Warnes et al., 2022*) and the ggstatsplot (*Patil, 2021*) R package were used for statistical analysis and graphical presentation.

## GSEA enrichment analysis

GSEA analysis was performed using R packages clusterProfiler (*Yu et al., 2012*) and msigdbr (*Liberzon et al., 2015*) to identify signaling pathways associated with high RFC3 expression.

## Analysis of immune infiltration and response to immunotherapy

The relationship between RFC3 expression and the tumor immune microenvironment (TIME) in DLBCL was investigated using immune infiltration analyses based on the GSE181063 dataset. The deconvo_tme function from the Immuno-Oncology Biological Research (IOBR) package (version 2.0.0) (*Zeng et al., 2024*) was used to compare the ImmuneScore, StromalScore, ESTIMATEScore, and Tumorpurity between the high and low RFC3 expression groups. These scores, derived from the ESTIMATE algorithm, quantify key components of the tumor microenvironment: ImmuneScore reflects the level of immune cell infiltration, StromalScore estimates the presence of stromal cells, and ESTIMATEScore combines both to represent the overall non-tumor content in the tumor tissue. Tumorpurity refers to the proportion of cancer cells in a tumor sample relative to non-cancerous cells, such as immune cells, stromal cells, and normal tissue. Additionally, immune cell infiltration differences were further assessed using single-sample

gene set enrichment analysis (ssGSEA) *via* the Gene Set Variation Analysis (GSVA) package (*Hanzelmann, Castelo & Guinney, 2013*) available on R software, using gene sets representing 28 tumor-infiltrating lymphocytes (TILs) obtained from the Tumor-Immune System Interactions Database (TISIDB) (*Barbie et al., 2009*). In addition, the correlation between RFC3 expression and immune-related genes was analyzed in DLBCL. Gene sets included immune inhibitor genes, immune stimulation genes, chemokine genes, Chemokine receptor genes, and immune checkpoint genes. These gene sets were selected as they are involved in a range of immunosuppressive, immune-activating, chemotactic, and immune checkpoint pathways. Spearman's correlation analysis was conducted for each gene in these categories. The immunotherapy response was predicted using the IMvigor210 cohort through the IMvigor210CoreBiologies R package (*Nickles & Bourgon, 2019*). This dataset includes transcriptomic and clinical data from patients with urothelial carcinoma treated with anti–PD-L1 therapy, allowing us to assess potential associations between RFC3 expression and immunotherapy outcomes.

### Drug-susceptibility analysis

The R package oncoPredict (*Maeser, Gruener & Huang, 2021*) was used to analyze the relationship between RFC3 expression and drug sensitivity in DLBCL. For this analysis, we obtained drug response data from the Genomics of Drug Sensitivity in Cancer (GDSC) data version 1 and 2 and the Cancer Therapeutics Response Portal, version 2 (CTRP2). These datasets include pharmacogenomic profiles of cancer cell lines exposed to a range of therapeutic agents for different RFC3 expression levels. The oncoPredict package was used to estimate the drug concentration required to inhibit 50% of the cell viability (IC50).

### Pan-cancer analysis of RFC3

The pan-cancer expression and prognostic value of RFC3 across different tumor types were investigated using the easyTCGA package (*Li, 2024*). The Cox proportional hazards model was used to evaluate the impact of RFC3 on OS, disease-specific survival (DSS), progression-free interval (PFI) and disease-free interval (DFI). Additionally, the R package TCGAplot (*Liao & Wang, 2023*) was employed to explore correlations between RFC3 expression and tumor mutational burden (TMB), as well as microsatellite instability (MSI), across multiple cancer types.

### Statistical analysis

All statistical analyses and graphical presentations of public data were performed using the R software version 4.2 (*R Core Team, 2022*). GraphPad Prism 9.0 (GraphPad Software, La Jolla, CA, USA) was used for statistical analysis and visualization of the IHC results. For all statistical tests, a *p*-value below 0.05 was deemed statistically significant. Significance levels were denoted as follows: *, $P < 0.05$. **, $P < 0.01$.***, $P < 0.001$. ****, $P < 0.0001$.

## RESULTS

### Up-regulation of RFC3 expression in DLBCL

The expression profile of RFC3 in DLBCL was systematically analyzed using multiple publicly available datasets and validated through IHC. Analysis of the GSE32018 dataset

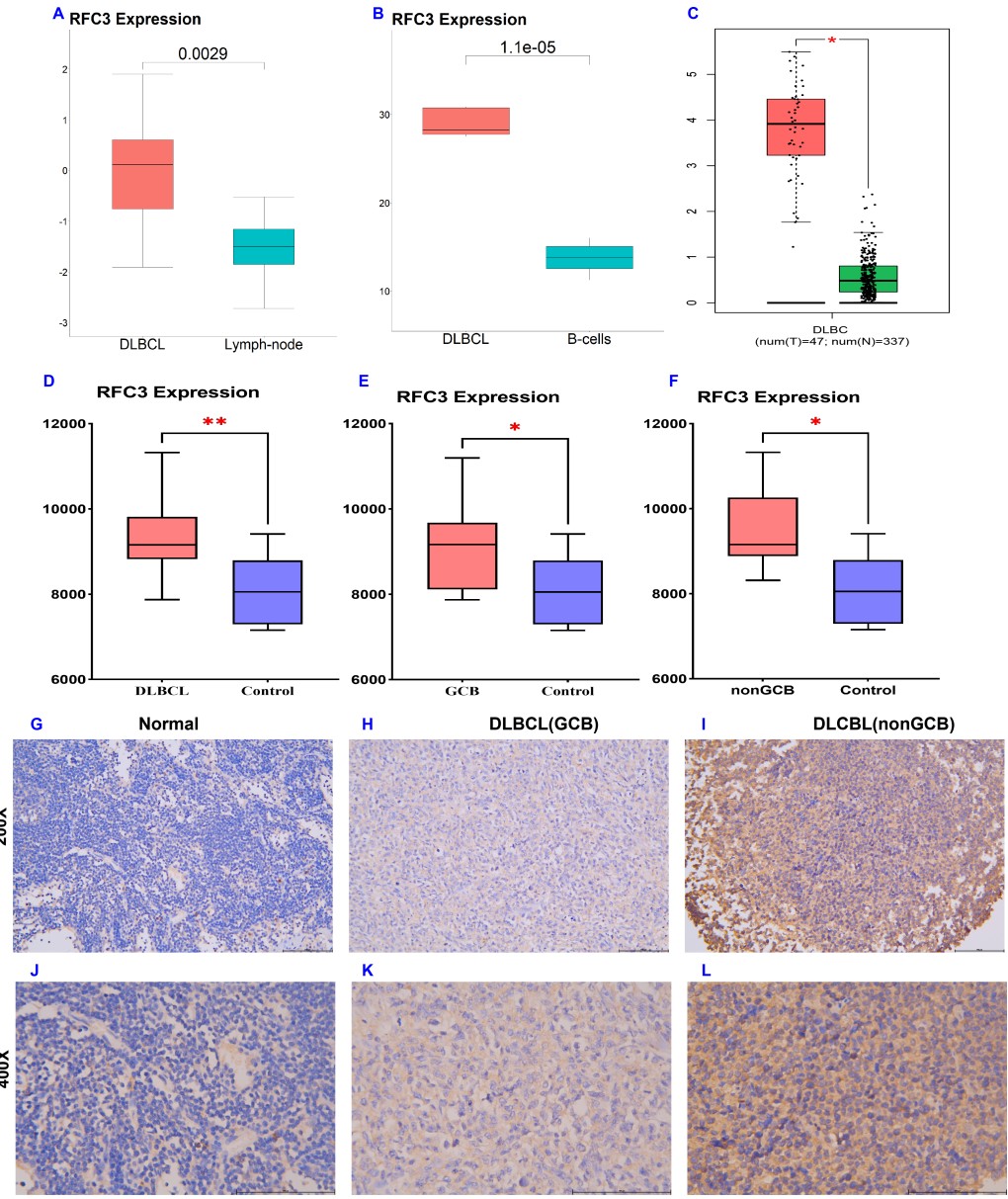

**Figure 1** **Increased RFC3 expression in patients with DLBCL.** Bioinformatics analysis showed that RFC3 expression was significantly increased in the public dataset GSE32018 (A), GSE12453 (B), and GEPIA (C). Immunohistochemical (IHC) validation of RFC3 overexpression in DLBCL tissues (D) and subtypes (E–F). Representative IHC images comparing RFC3 expression in DLBCL (H–I & K–L) and normal lymph nodes (G & J).

revealed that RFC3 was significantly overexpressed in DLBCL compared to normal lymphoid tissues ($P = 0.0029$, Fig. 1A). Similarly, analysis of the GSE112453 dataset showed higher RFC3 expression in DLBCL relative to B-cells ($P = 1.1e-05$, Fig. 1B). Consistent with these findings, data from the GEPIA database demonstrated significantly

elevated RFC3 expression in TCGA-DLBC samples compared to normal tissue ($P < 0.05$, Fig. 1C).

To validate these results at the protein level, IHC was performed on clinical samples, including DLBCL tissues and normal lymph nodes. The analysis confirmed significantly elevated RFC3 expression in DLBCL samples compared to normal lymph node tissues ($P < 0.01$, Fig. 1D). Notably, overexpression was observed in both germinal center B-cell-like (GCB) ($P < 0.05$, Fig. 1E) and non-GCB subtypes of DLBCL ($P < 0.05$, Fig. 1F), but there was no difference in RFC3 expression between these two subtypes. Representative immunohistochemical images are shown in Figs. 1G to 1L, indicating that RFC3 dysregulation is a common feature across molecular subtypes of DLBCL. Collectively, these results establish RFC3 as a consistently overexpressed gene in DLBCL.

### High RFC3 expression indicates poor prognosis in DLBCL

Survival analysis revealed that high RFC3 expression was significantly associated with worse overall survival (OS) in the GSE181063 ($P = 0.0022$, Fig. 2A), GSE10846 ($P = 0.0053$, Fig. 2B), and GSE32918 ($P = 0.0071$, Fig. 2C) cohorts. Additionally, in the GSE31312 cohort, both OS ($P = 0.014$, Fig. 2D) and progression free survival (PFS) ($P = 0.029$, Fig. 2E) were significantly shorter in DLBCL patients with high RFC3 expression compared to those with low expression. These findings indicate that elevated RFC3 expression could potentially be used as a negative prognostic marker in DLBCL.

### Elevated RFC3 expression correlates with aggressive clinical features in DLBCL

The results of the association between RFC3 expression levels and clinical characteristics of DLBCL are shown in Table 2. High RFC3 expression was significantly correlated with several known unfavorable prognostic factors, including ECOG > 1 ($P = 0.03$, Fig. 3A), raised LDH levels ($P = 3.32e-03$, Fig. 3B), Ann Arbor stages III–IV ($P = 0.02$, Fig. 3C), high IPI risk (>2) ($P = 4.97e-07$, Fig. 3D), extranodal involvement in more than two sites ($P = 0.04$, Fig. 3E), and MHG subtype ($P = 1.12e-13$, Fig. 3F). However, no significant association was observed between RFC3 expression and demographic variables such as age and gender.

### RFC3 plays a role in various tumor-related signaling pathways in DLBCL

Enrichment analysis revealed that RFC3 expression was significantly correlated with multiple signaling pathways in DLBCL (Table 3). Higher RFC3 levels were notably associated with key tumor-related pathways (Figs. 4A & 4B), including DNA repair, E2F transcription factor targets (E2F_TARGETS), G2/M cell cycle checkpoint (G2M_CHECKPOINT), mechanistic target of rapamycin complex 1 (mTORC1) signaling, MYC proto-oncogene transcription factor targets V1 & V2 (MYC Targets V1 and MYC Targets V2), oxidative phosphorylatio (OXPHOS), unfolded protein response (UPR).

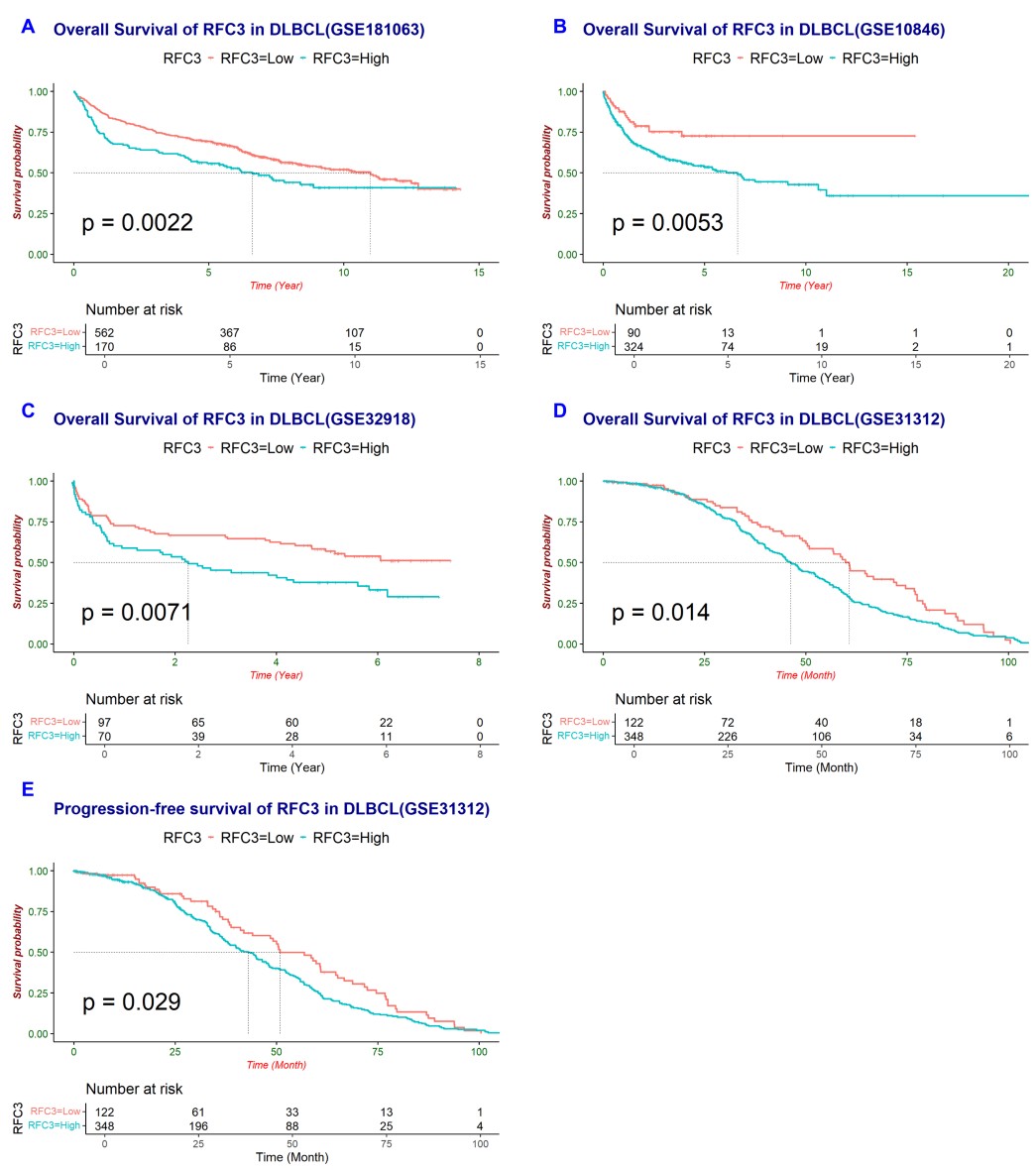

**Figure 2  RFC3 is associated with poor prognosis in DLBCL.** Survival analysis indicates that higher RFC3 expression is associated with poorer overall survival (OS) in DLBCL patients across four independent cohorts: GSE181063 (A), GSE10846 (B), GSE32918 (C), and GSE31312 (D). Moreover, increased RFC3 expression is also associated with shorter PFS (E).

## The expression of RFC3 is significantly associated with immune infiltration and various immune factors, indicating a differential response to immunotherapy

Elevated RFC3 expression was correlated with a significant decrease in the ESTIMATEScores (P < 2e−16), ImmuneScore (P < 2e−16), and StromalScore (P = 9.1e−16), along with a marked increase in TumorPurity (P < 2e−16) (Fig. 5A). These results suggest the presence of a "cold" immune microenvironment characterized by a reduction in immune

**Table 2  Correlation between RFC3 expression and clinicopathological features in DLBCL.**

| Characteristic | Low expression of RFC3 N (%) | High expression of RFC3 N (%) | χ2 | P value |
|---|---|---|---|---|
| COO | | | 63.37 | <0.001* |
|   UNC | 116 (20.6) | 10 (5.9) | | |
|   ABC | 171 (30.4) | 42 (24.7) | | |
|   GCB | 256 (45.6) | 86 (50.6) | | |
|   MHG | 19 (3.4) | 32 (18.8) | | |
| Age | | | 2.45 | 0.117 |
|   ≦60y | 174 (31) | 42 (24.7) | | |
|   >60y | 388 (69) | 128 (75.3) | | |
| Sex | | | 0.11 | 0.745 |
|   Female | 273 (48.6) | 85 (50) | | |
|   Male | 289 (51.4) | 85 (50) | | |
| ECOG | | | 4.85 | 0.028* |
|   ECOG ≦1 | 425 (82.2) | 115 (74.2) | | |
|   ECOG >1 | 92 (17.8) | 40 (25.8) | | |
| Stage | | | 5.46 | 0.019* |
|   Stage I–II | 202 (40.6) | 45 (30.0) | | |
|   Stage III–IV | 296 (59.4) | 105 (70.0) | | |
| IPI | | | 25.27 | <0.001* |
|   Low-Risk | 270 (62.8) | 45 (37.2) | | |
|   High-Risk | 160 (37.2) | 76 (62.8) | | |
| LDH | | | 8.62 | 0.009* |
|   Normal | 239 (47.1) | 50 33.6) | | |
|   Raised | 268 (52.9) | 99 (66.4) | | |
| Bs | | | 2.76 | 0.096 |
|   Not | 359 (63.9) | 96 (56.8) | | |
|   Have | 203 (36.1) | 73 (43.2) | | |
| Extranodal | | | 4.28 | 0.038* |
|   <2 | 189 (83.6) | 50 (72.5) | | |
|   ≥2 | 37 (16.4) | 19 (27.5) | | |

**Notes.**
*$P < 0.05$.
Pearson χ2 test.
RFC3, Replication Factor C Subunit 3.

cell populations and an abundance of tumor cells. Immune cell profiling utilizing ssGSEA analysis demonstrated a positive correlation between RFC3 expression and pro-tumorigenic immune subsets, such as activated CD4 T cells, memory B cells, activated B cells, immature B cells, central memory CD8 T cells, gamma delta T cells, type 2 T helper cells, CD56 bright natural killer cells, natural killer cells, eosinophils, type 1 T helper cells, monocytes, effector memory CD4 T cells, and plasmacytoid dendritic cells. Conversely, RFC3 expression was inversely associated with activated dendritic cells, T follicular helper cells, mast cells, natural killer T cells, and neutrophils (Fig. 5B).

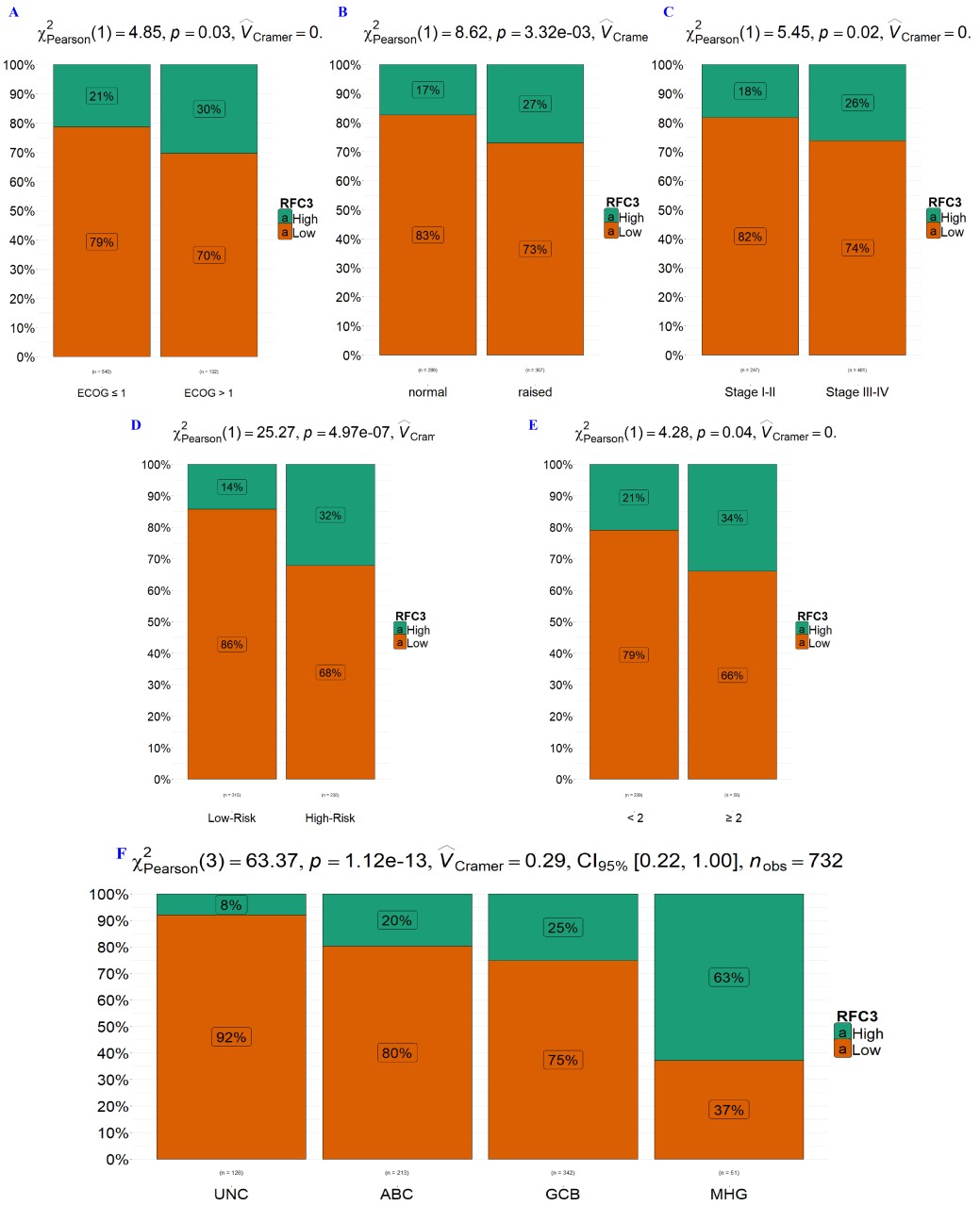

**Figure 3  High RFC3 expression correlates with unfavorable clinical features in DLBCL.** Elevated expression of RFC3 is significantly associated with the following factors: (A) Eastern Cooperative Oncology Group Performance Status (ECOG) > 1, (B) raised lactate dehydrogenase (LDH) levels, (C) advanced stage (Stage III-IV), (D) high-risk International Prognostic Index (IPI) score, (E) involvement of more than two extranodal sites (≥ 2), and (F) unfavorable cell-of-origin (COO) subtypes.

Analyses using the TISIDB revealed strong positive correlations between RFC3 expression and immune stimulator genes such as MICB, CD70, CD48, *etc.*, but negatively correlated with TNFSF14, RAET1E, TNFRSF25, TNFRSF4 and multiple co-stimulatory molecules (Fig. 6A). Among the immune inhibitor genes RFC3 positively correlated

**Table 3  Results of the enrichment analysis of RFC3 expression in DLBCL.**

|  | setSize | NES | p value | q value |
|---|---|---|---|---|
| HALLMARK_E2F_TARGETS | 182 | 4.1 | 1.00E−10 | 2.42E−10 |
| HALLMARK_MYC_TARGETS_V1 | 185 | 4.01 | 1.00E−10 | 2.42E−10 |
| HALLMARK_G2M_CHECKPOINT | 181 | 3.79 | 1.00E−10 | 2.42E−10 |
| HALLMARK_OXIDATIVE_PHOSPHORYLATION | 179 | 3.5 | 1.00E−10 | 2.42E−10 |
| HALLMARK_MTORC1_SIGNALING | 190 | 3.3 | 1.00E−10 | 2.42E−10 |
| HALLMARK_DNA_REPAIR | 137 | 3.3 | 1.00E−10 | 2.42E−10 |
| HALLMARK_MYC_TARGETS_V2 | 56 | 3.27 | 1.00E−10 | 2.42E−10 |
| HALLMARK_UNFOLDED_PROTEIN_RESPONSE | 106 | 3.04 | 1.00E−10 | 2.42E−10 |
| HALLMARK_MITOTIC_SPINDLE | 194 | 2.97 | 1.00E−10 | 2.42E−10 |
| HALLMARK_FATTY_ACID_METABOLISM | 151 | 2.38 | 4.08E−10 | 8.98E−10 |
| HALLMARK_REACTIVE_OXYGEN_SPECIES_PATHWAY | 44 | 2.29 | 3.28E−06 | 5.68E−06 |
| HALLMARK_PROTEIN_SECRETION | 95 | 2.21 | 1.46E−06 | 2.71E−06 |
| HALLMARK_ADIPOGENESIS | 186 | 2.08 | 8.89E−08 | 1.79E−07 |
| HALLMARK_PEROXISOME | 99 | 2.07 | 2.70E−05 | 4.09E−05 |
| HALLMARK_CHOLESTEROL_HOMEOSTASIS | 70 | 1.98 | 9.92E−05 | 0.000123 |
| HALLMARK_PI3K_AKT_MTOR_SIGNALING | 103 | 1.93 | 7.31E−05 | 9.83E−05 |
| HALLMARK_GLYCOLYSIS | 193 | 1.89 | 9.78E−06 | 1.58E−05 |
| HALLMARK_UV_RESPONSE_UP | 151 | 1.88 | 4.12E−05 | 5.86E−05 |
| HALLMARK_ANDROGEN_RESPONSE | 96 | 1.65 | 0.006425 | 0.006763 |
| HALLMARK_APOPTOSIS | 158 | 1.49 | 0.013551 | 0.012619 |
| HALLMARK_P53_PATHWAY | 187 | 1.48 | 0.006792 | 0.006851 |
| HALLMARK_INFLAMMATORY_RESPONSE | 197 | −1.37 | 0.007781 | 0.007535 |
| HALLMARK_EPITHELIAL_MESENCHYMAL_TRANSITION | 195 | −1.67 | 0.000212 | 0.000245 |
| HALLMARK_MYOGENESIS | 194 | −1.72 | 0.000102 | 0.000123 |
| HALLMARK_COAGULATION | 135 | −1.74 | 0.000233 | 0.000257 |
| HALLMARK_KRAS_SIGNALING_DN | 187 | −2.36 | 1.00E−10 | 2.42E−10 |

with PDCD1LG2, but negative associations with KIR2DL1, TIGIT, and others (Fig. 6B). RFC3 shows a positive correlation with the chemokine CX3CL1, while displaying negative correlations with multiple T-cell trafficking-related chemokines (*e.g.*, CCL26, CXCL14, CCL16, CCL24) (Fig. 6C). Among chemokine receptors, RFC3 is positively linked to CXCR5, CCR10, CXCR4 but negatively associated with CXCR1, CCR6, CCR8, and others (Fig. 6D).

RFC3 was positively associated with the immune checkpoint gene PDCD1LG2 but inversely correlated with CTLA4, PDCD1, and TIGIT (Fig. 6E), highlighting its dual role in modulating immune activation and exhaustion. This suggests RFC3 may influence immune cell migration by modulating chemokine signaling pathways. Collectively, these findings position RFC3 as a key orchestrator of immune evasion in DLBCL, attenuating cytotoxic responses while promoting myeloid-driven immunosuppression. Furthermore, analysis using the IMvigor210CoreBiologies R package revealed that patients with high

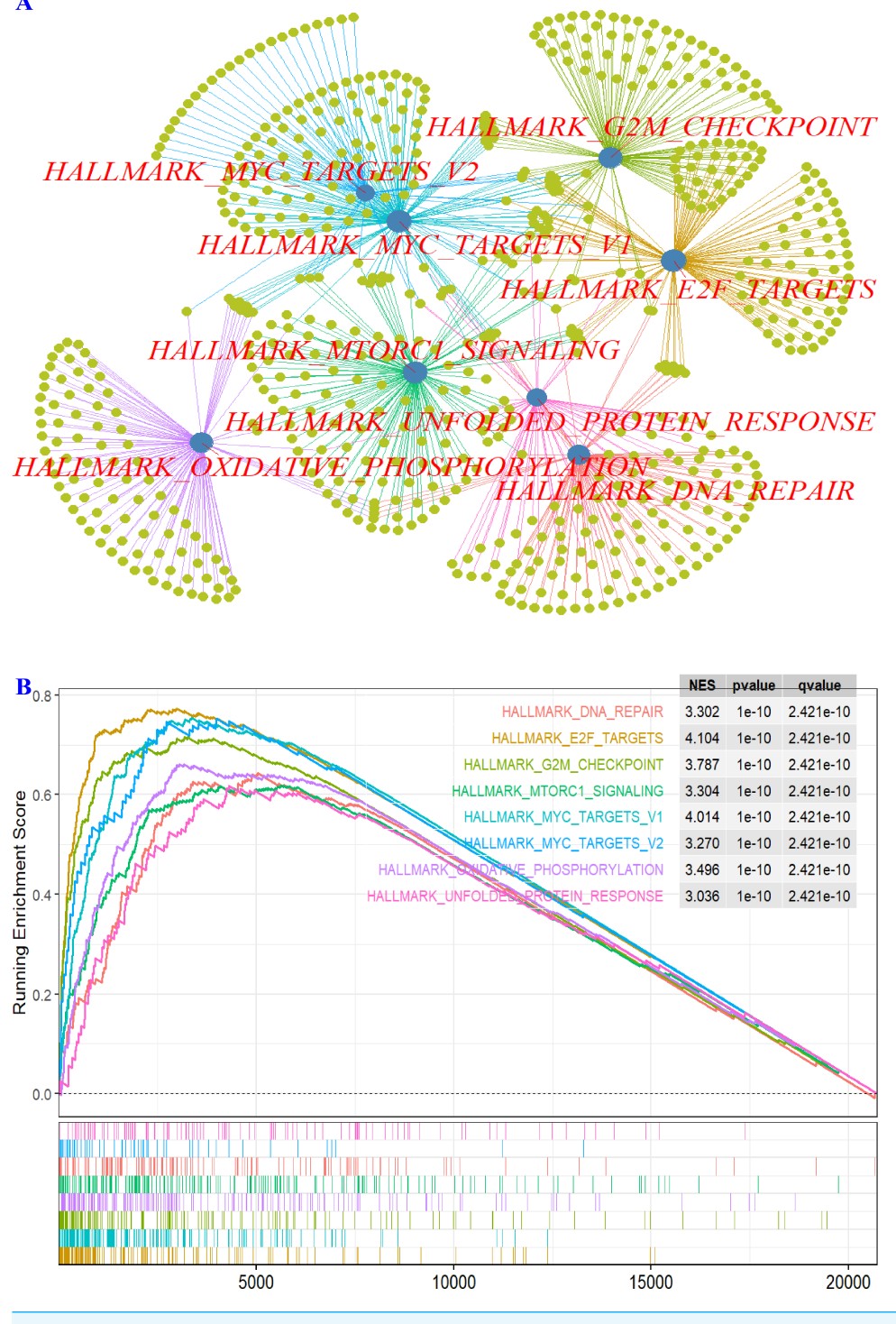

**Figure 4** (A–B) **Enrichment analysis.** The top eight significantly enriched tumor-related pathways are presented.

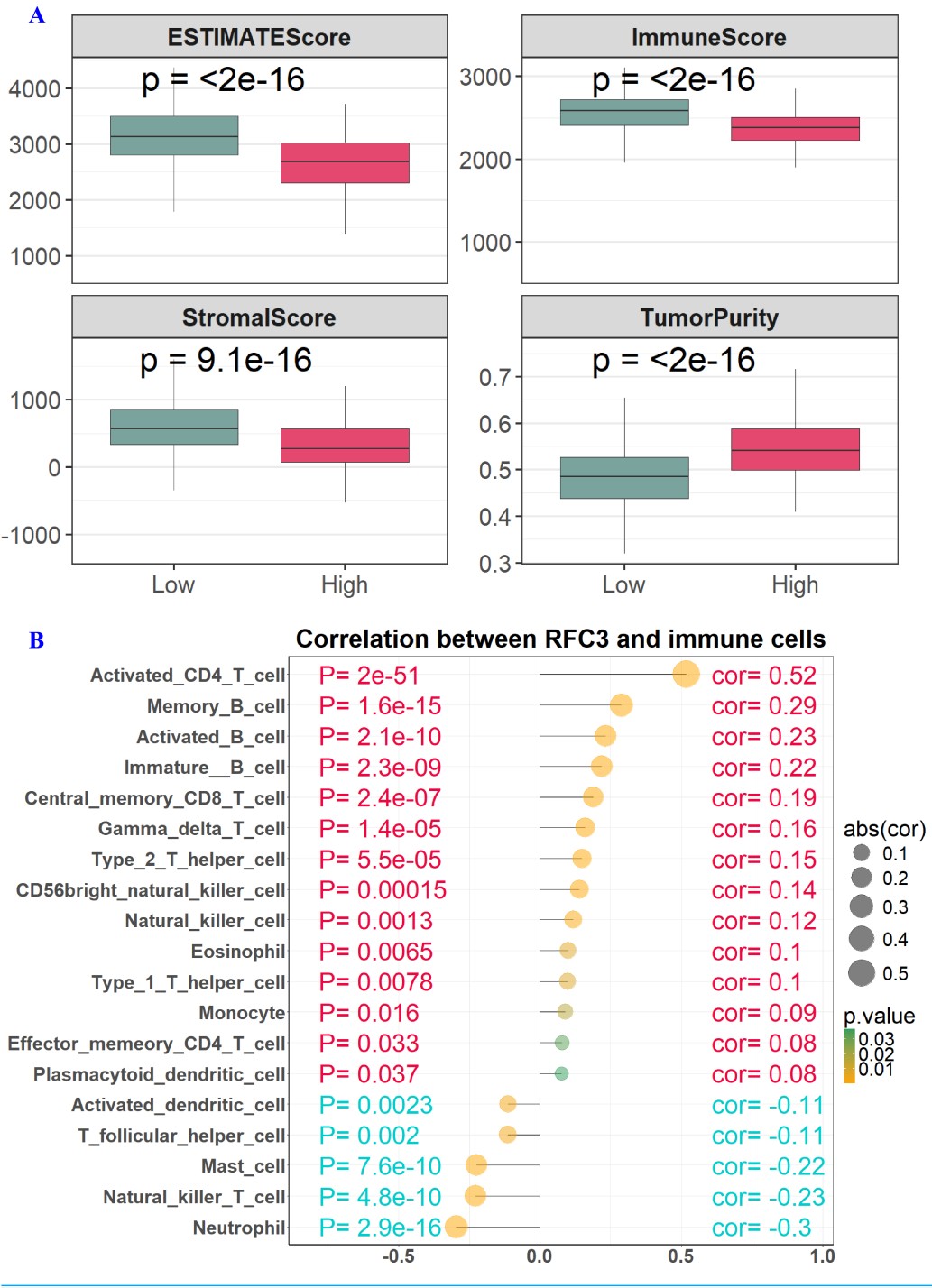

**Figure 5** **The association between RFC3 expression and the tumor immune microenvironment.** The RFC3 high expression group had significantly lower ESTIMATEScore, ImmuneScore, StromalScore and higher TumorPurity (A) than the RFC3 low expression group. Application of ssGSEA analysis showed a significant correlation between RFC3 expression and immune cell infiltration (B).

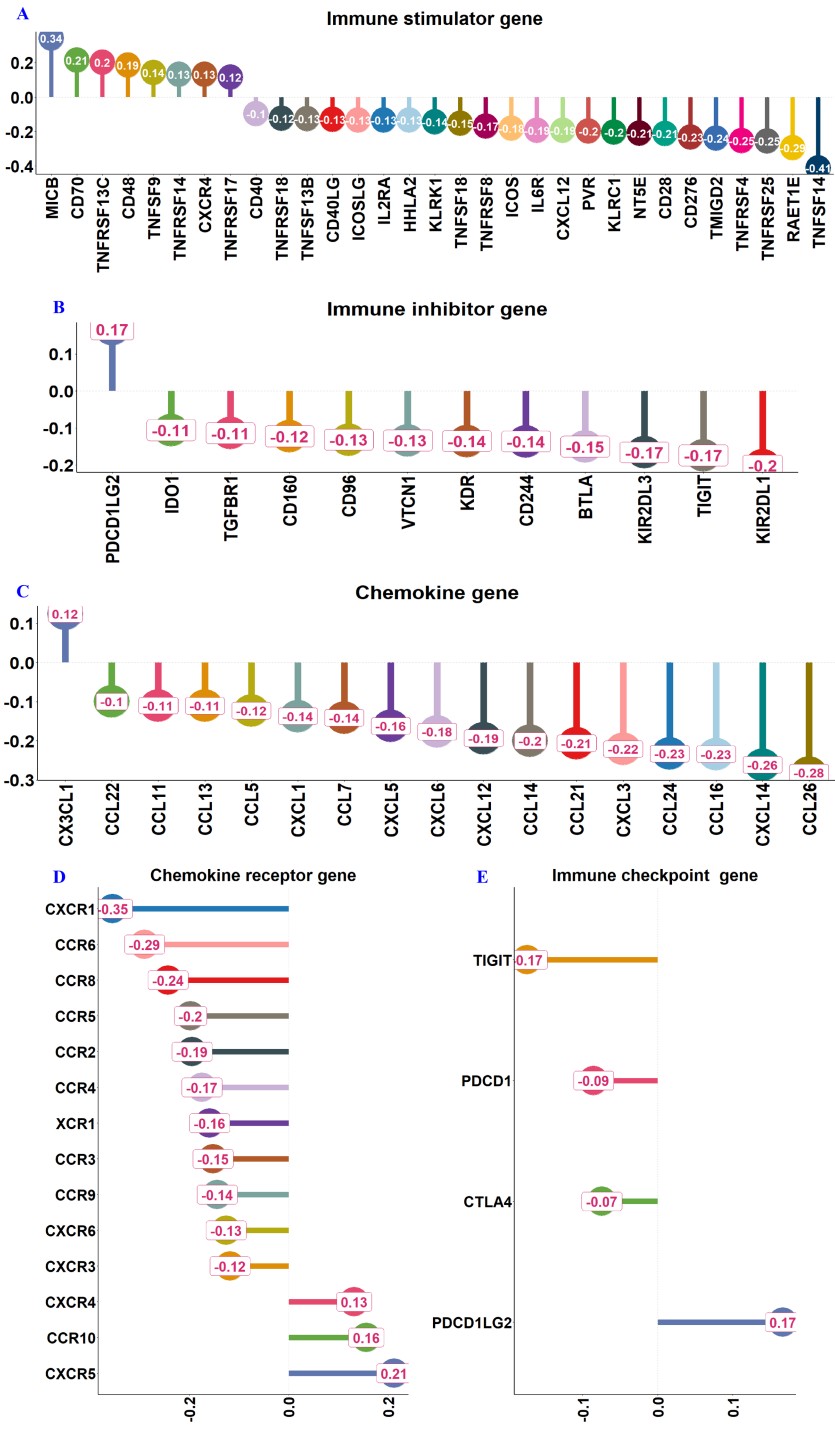

**Figure 6** **RFC3 expression correlates with immune-related factors.** (A) Correlation between RFC3 expression and immune stimulator genes. (B) Association of RFC3 expression with Immune inhibitor genes. (C) Relationship between RFC3 expression and chemokine genes. (D) Correlation of RFC3 expression with chemokine receptor genes. (E) Association of RFC3 with immune checkpoint genes.

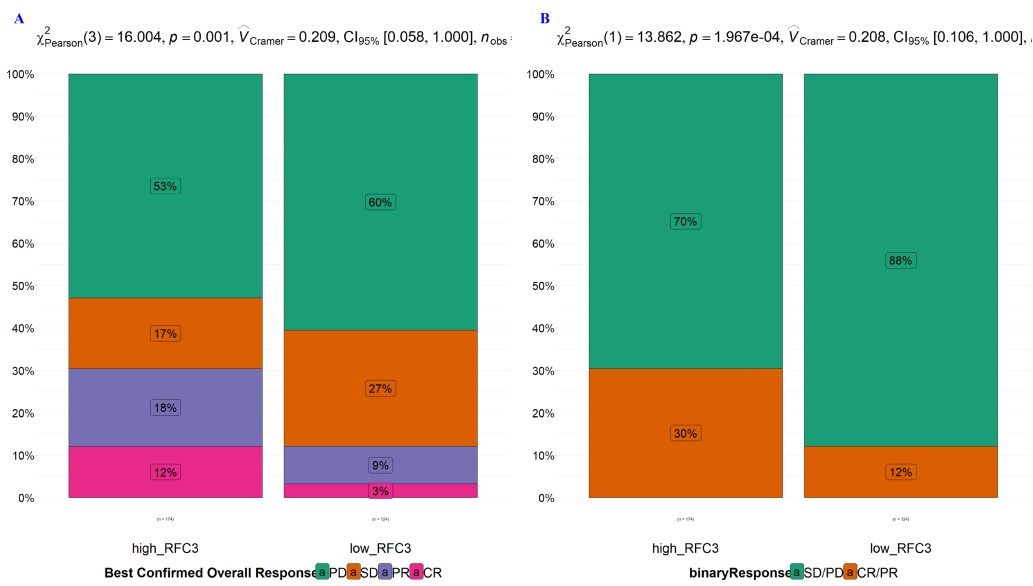

**Figure 7** (A–B) High RFC3 expression predicts a better response to immunotherapy.

RFC3 expression exhibited significantly higher rates of complete or partial response to chemotherapy (CR/PR: 30% *vs.* 12%, $p < 0.01$) (Figs. 7A & 7B).

## The expression of RFC3 demonstrated a significant correlation with drug sensitivity

Figures 8A to 8C present a graphical representation of drugs showing a significant association with RFC3 expression (correlation coefficient >0.5 and $p < 0.01$). The analysis revealed a positive correlation between RFC3 expression and the concentrations of cyclophosphamide and bleomycin, both commonly used in lymphoma treatment (Figs. 9A & 9B), suggesting that elevated RFC3 levels may contribute to resistance against these agents. In contrast, concentrations of newer lymphoma therapies—such as venotoclax, lenalidomide, decitabine, and ibrutinib were negatively correlated with RFC3 expression (Figs. 9C–9F). This inverse relationship indicates that these drugs may be more effective in patients with high RFC3 expression, highlighting their potential benefit in lymphoma treatment.

## RFC3 exhibits abnormal expression patterns across a range of tumors and is linked to prognostic outcomes

A comprehensive pan-cancer analysis of RFC3 expression, using matched normal tissues as controls, revealed significant dysregulation across various cancer types. Notable increases in RFC3 levels were observed in cancers originating from epithelial tissues, such as bladder urothelial carcinoma (BLCA), breast invasive carcinoma (BRCA), and cervical squamous cell carcinoma (CESC). Additionally, elevated RFC3 expression was found in gastrointestinal cancers such as cholangiocarcinoma (CHOL), colon Adenocarcinoma (COAD), esophageal carcinoma (ESCA), and Stomach adenocarcinoma (STAD). Increased

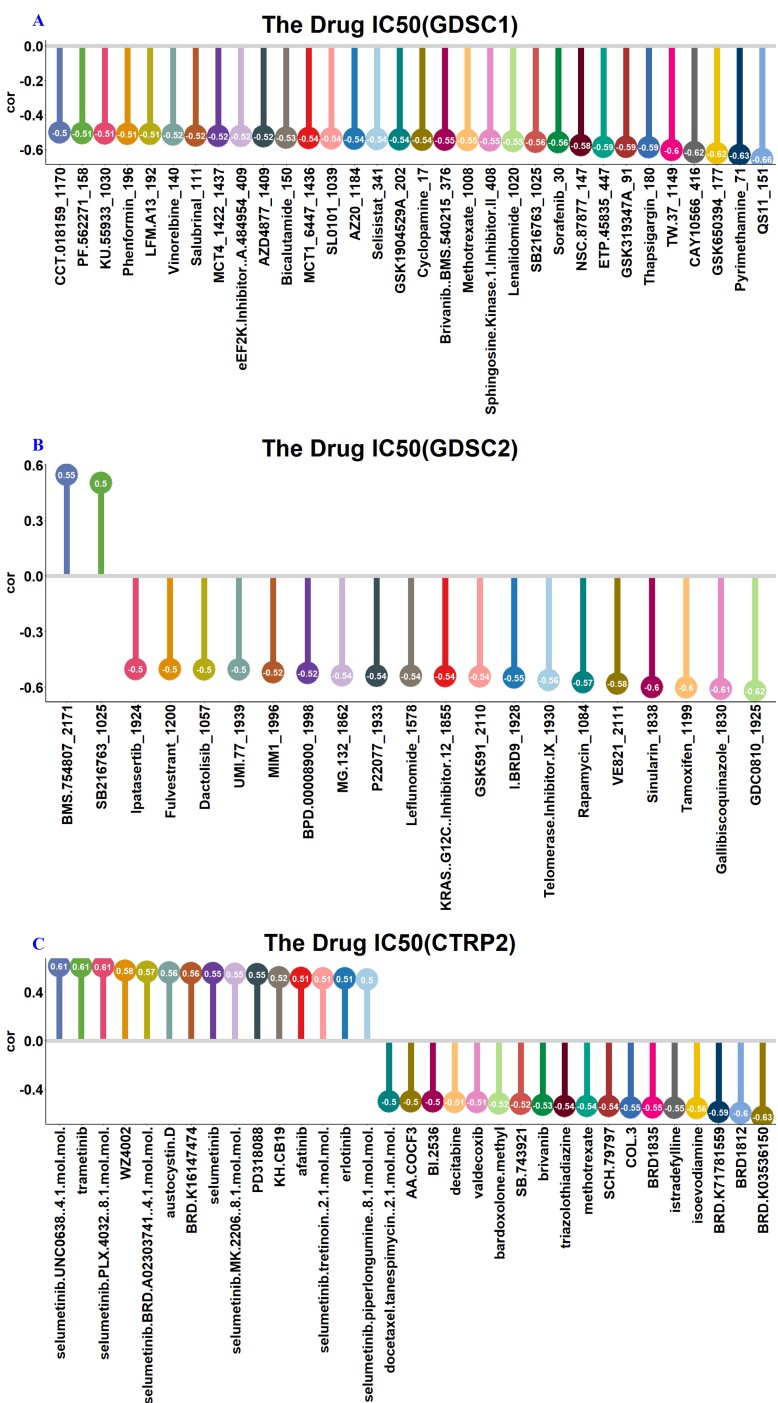

**Figure 8 The expression of RFC3 is correlated with sensitivity of various drugs.** Analysis between RFC3 expression and drug sensitivity was performed based on GDSC1 (A), GDSC2 (B) and CTRP2 (C) datasets, respectively.

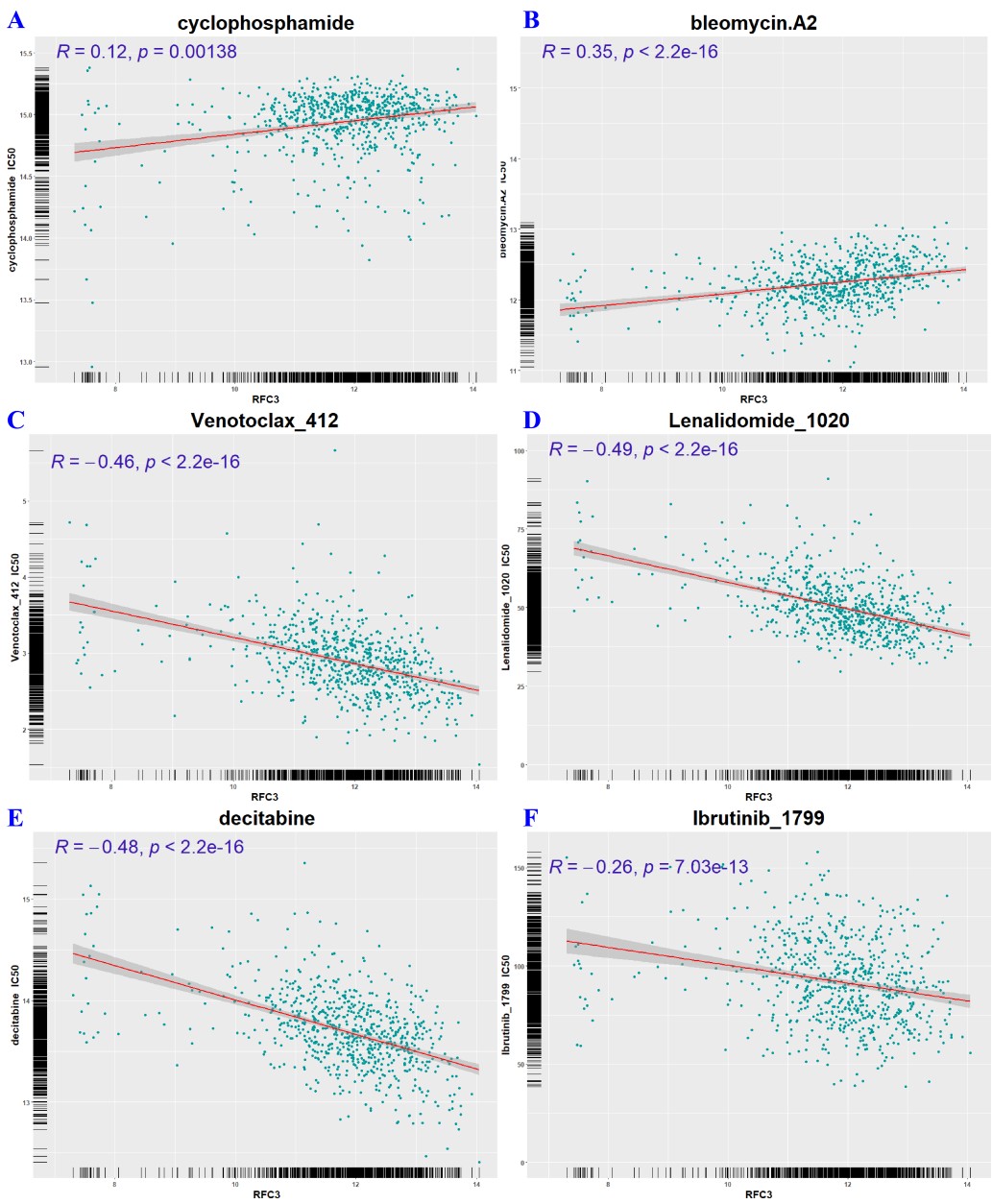

**Figure 9  RFC3 expression acts as a predictive biomarker for chemotherapeutic response.** High RFC3 levels correlate with reduced sensitivity to cyclophosphamide (A) and bleomycin (B), whereas they increase susceptibility to venetoclax (C), lenalidomide (D), decitabine (E), and ibrutinib (F).

RFC3 levels were also noted in neurological cancers (glioblastoma multiforme (GBM), lower grade glioma (LGG)), thoracic cancers (lung adenocarcinoma (LUAD), lung squamous cell carcinoma (LUSC)), and reproductive system cancers (ovarian serous cystadenocarcinoma (OV), uterine corpus endometrial carcinoma (UCEC), Uterine carcinosarcoma (UCS)). Other cancers exhibiting heightened RFC3 levels included head and neck squamous cell carcinoma (HNSC), liver hepatocellular carcinoma

(LIHC), pancreatic adenocarcinoma (PAAD), prostate adenocarcinoma (PRAD), rectum adenocarcinoma (READ), sarcoma (SARC), skin cutaneous melanoma (SKCM), and testicular germ cell tumors (TGCT). Conversely, RFC3 expression was reduced in kidney chromophobe (KICH), acute myeloid leukemia (LAML), thyroid carcinoma (THCA), and thymoma (THYM), indicating that its regulatory functions may differ by tissue type (Fig. 10A).

Survival analysis indicates that the aberrant expression of RFC3 is significantly associated with various survival outcomes across a range of tumor types (Fig. 10B). Specifically, elevated RFC3 expression correlates with diminished OS in malignancies such as ACC and CESC. Furthermore, in tumors for instance ACC, CESC, KICH and KIRC, increased RFC3 expression is significantly linked to a reduction in DSS. Additionally, in malignancies *e.g.*, ACC, BLCA, BRCA, higher levels of RFC3 are significantly correlated with PFI. Moreover, in ACC, BRCA, CESC, COAD, and other cancers, elevated RFC3 expression is associated with an unfavorable DFI.

Additionally, the expression of RFC3 exhibits a positive correlation with TMB across various cancer types, including BLCA, KICH, LGG, LUAD, PRAD, and UCEC (Fig. 10C). Furthermore, RFC3 expression positively correlated with MSI in KIRC, READ, SARC, STAD, and UCEC, while exhibiting a negative correlation with MSI in DLBCL (Fig. 10D). These findings suggest that RFC3 may play distinct roles in genomic instability and the tumor immune microenvironment across several cancer types.

## DISCUSSION

Patients diagnosed with diffuse large B-cell lymphoma (DLBCL) display diverse survival outcomes, reflecting the disease's inherent heterogeneity (*Shi et al., 2019*). Recent investigations into the genetics of DLBCL have significantly advanced molecular subtyping and prognostic modeling. For instance, *Lacy et al. (2020)* identified five distinct molecular subtypes demonstrating their prognostic significance and clinical applicability in patients treated with R-CHOP. *Cui & Leng (2023)* developed an eight-gene risk model related to glycolysis, which predicted a poor prognosis and an association with immune cell infiltration. *Chen et al. (2021)* proposed eight gene signatures associated with ferroptosis including ZEB1 and NFE2L2 which effectively stratified DLBCL patients into different risk categories and identified potential therapeutic targets. Gene-expression profiling (GEP) enhances the subtyping of diffuse large B-cell lymphoma (DLBCL), elucidates the roles of the tumor microenvironment (TME), and provides valuable molecular insights. The integration of multi-omics approaches enhances diagnostic accuracy, prognostic assessments, and therapeutic strategies (*De Groot et al., 2022*). Single-cell RNA sequencing of DLBCL has revealed considerable tumor heterogeneity and elucidated the mechanisms of T-cell exhaustion associated with TIM3 and TIGIT (*Ye et al., 2022*). These findings may contribute to the development of targeted therapeutic approaches.

Previous research has demonstrated that RFC3 is markedly overexpressed in various malignant tumors, RFC3 demonstrates elevated expression levels in head and neck squamous cell carcinoma (HNSCC), correlating with tumor progression, reduced survival

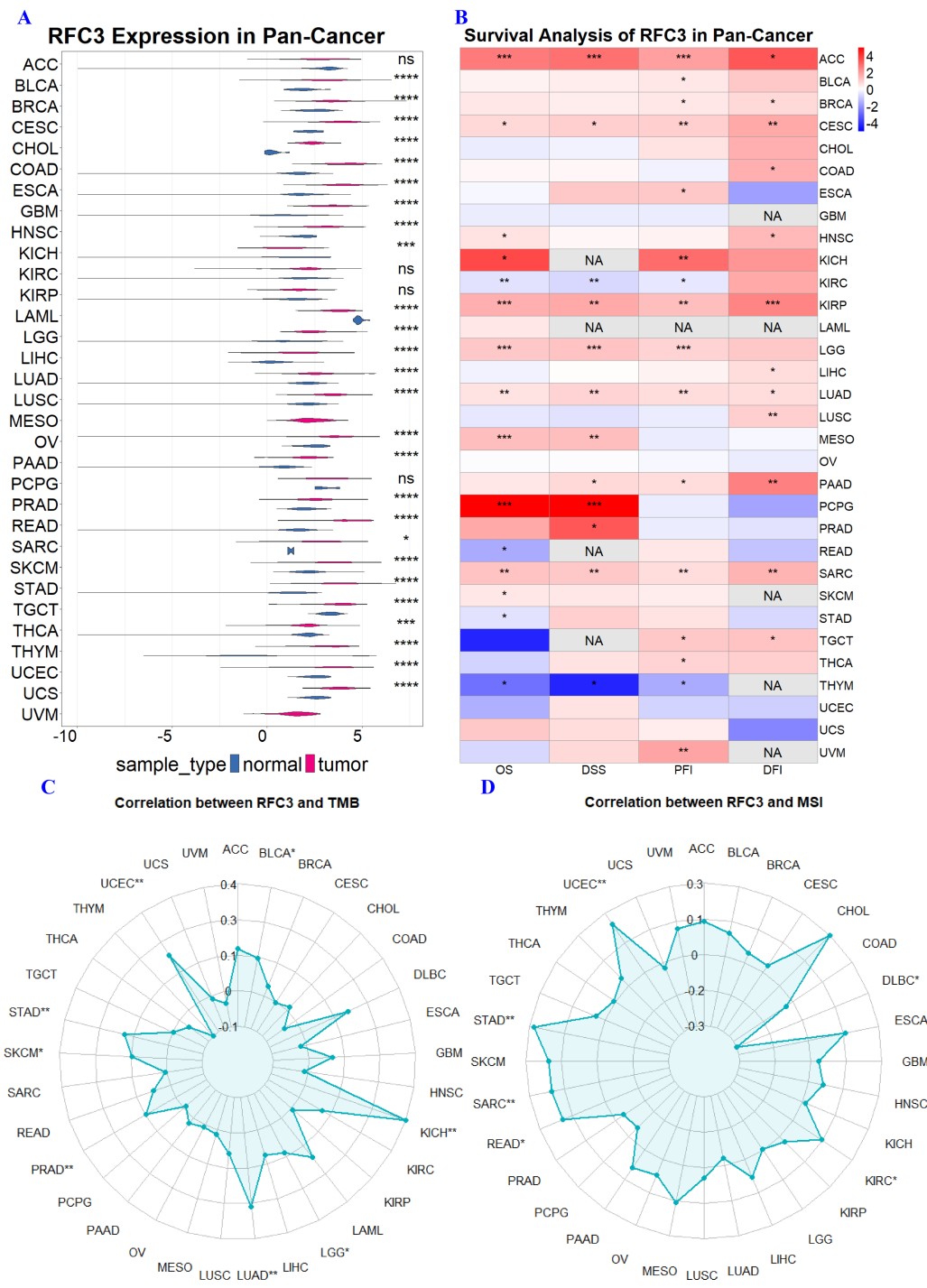

**Figure 10** Pan-cancer analysis of RFC3 in TCGA tumors reveals significantly differential expression of RFC3 (A), and is associated with multiple survival indicators, including overall survival (OS), disease-specific survival (DSS), progression-free interval (PFI) and disease-free interval (DFI) (B) across various cancer types. Furthermore, RFC3 expression demonstrates significant correlations with both tumor mutational burden (TMB) (C) and microsatellite instability (MSI) (D) across diverse TCGA malignancies.

rates, and the infiltration of immune cells. Its involvement in the regulation of DNA repair processes and oncogenic signaling pathways suggests that RFC3 could function as a significant prognostic biomarker and a prospective target for therapeutic strategies (*Avs et al., 2023*). RFC3 plays a significant role in the progression of triple-negative breast cancer (TNBC) by promoting the process of epithelial-mesenchymal transition (EMT). Research indicates that inhibiting RFC3 can lead to a reduction in metastasis and tumor growth. Furthermore, the overexpression of RFC3 is associated with a decline in patient prognosis, underscoring its potential utility as both a prognostic biomarker and a target for therapeutic strategies (*He et al., 2017*).

Our study revealed that RFC3 is significantly overexpressed in DLBCL. Elevated RFC3 expression was observed in both germinal center B-cell (GCB) and non-GCB subtypes, markedly higher than in normal controls. Furthermore, increased RFC3 expression was significantly associated with worse OS and PFS in DLBCL patients. Moreover we found that elevated RFC3 expression significantly correlates with several adverse prognostic factors in DLBCL, including age over 60, increased LDH levels, high IPI score, and advanced Ann Arbor stages (Stage III–IV), suggesting a role for RFC3 in the biological mechanisms underlying these features. Research on gene expression profiling conducted on a cohort of 928 patients has demonstrated that the molecular high-grade (MHG) subtype of DLBCL represents a biologically distinct subgroup associated with a poor prognosis. The study revealed that MHG exhibits a proliferative, centroblast-like phenotype and is linked to significantly reduced progression-free survival following treatment with R-CHOP. The identification of MHG enhances the characterization of the poor-prognosis category, thereby facilitating the development of intensified or targeted therapeutic strategies (*Sha et al., 2019*). Our analysis showed a significant association between RFC3 expression and the aggressive MHG subtype. Additionally, pan-cancer analysis indicated that RFC3 is overexpressed across multiple tumor types. Overexpression of RFC3 was also significantly associated with prognosis, highlighting its potential as a therapeutic target in oncology.

The emergence and progression of tumors are closely associated with the dysregulation of various tumor signaling pathways, uncontrolled cell proliferation is one of the key indicators of tumor aggressiveness. In patients with estrogen receptor-positive (ER-positive) breast cancer, MYC target scores were significantly correlated with tumor aggressiveness and worse survival outcomes (*Schulze et al., 2020*). The E2F pathway score served as a novel predictive marker for therapeutic response in ER+/HER2- breast cancer (*Oshi et al., 2020*). In pancreatic cancer, the G2/M checkpoint pathway was linked to treatment response and survival outcomes (*Oshi et al., 2021*). The upregulation of genes associated with DNA repair was found to contribute to an increased incidence of early recurrence in childhood precursor B-cell acute lymphoblastic leukemia (B-ALL) (*Albaqami et al., 2024*). Our pathway enrichment analysis revealed that RFC3 overexpression was associated with critical pathways involved in cell proliferation, including the G2M checkpoint, oxidative phosphorylation, E2F targets, MTORC1 signaling, MYC targets, and DNA repair. Our findings indicate that these pathways play vital roles in cancer progression, highlighting the essential function of RFC3 in sustaining the malignant phenotype of DLBCL.

The tumor immune microenvironment also plays a very important role in tumor progression. Studies have shown that in CRC (*Lin et al., 2024*), HCC (*Sun et al., 2023*), and NSCLC (*Chen et al., 2024*), high-risk subtypes with poor prognoses exhibit lower Stromalscores, Immunescores, and ESTIMATEscores, alongside higher TumorPurity. These findings suggest that patients with these malignancies treated with immune checkpoint inhibitors (ICI) may experience poorer outcomes. Similarly, the tumor microenvironment plays a crucial role in the initiation and progression of DLBCL (*Cioroianu et al., 2019*), with patient prognosis closely linked to the characteristics of the tumor immune microenvironment (*Ciavarella et al., 2019*). The infiltration of diverse immune cell types within the DLBCL microenvironment contributes significantly to disease heterogeneity, with the composition and abundance of these cells correlating with clinical outcomes. Mast cell infiltration has been identified as a favorable prognostic marker, while higher levels of CD8+ and CD4+ T cells predict better survival in DLBCL (*Kusano et al., 2017*; *Qi et al., 2022*). Conversely, lower NK cell counts are associated with shorter PFS and OS (*Lin et al., 2023*). An imbalance of immune cells may disrupt the immune microenvironment, promoting immunosuppression in DLBCL.

Moreover, chemokines and their receptors play critical roles in cancer initiation, progression, and metastasis, while immune checkpoint inhibitors have shown efficacy in certain refractory hematological malignancies (*Thanarajasingam, Thanarajasingam & Ansell, 2016*). In line with this, our study found significant differences in the tumor microenvironment between high and low RFC3 expression groups. The Tumorpurity was significantly higher in the RFC3-high group, whereas the StromalScore, ImmuneScore, and ESTIMATEScore were markedly lower. Immune cell infiltration analysis revealed distinct patterns: RFC3-high expression positively correlated with activated CD4+ T cells, memory B cells, activated B cells, and type 2 helper T cells, but negatively correlated with key immune effectors, including T follicular helper cells, mast cells, neutrophils, and natural killer T cells. This immune evasion mechanism may underlie the poor prognosis associated with elevated RFC3 expression.

Furthermore, immune-related factors showed significant correlations with RFC3 expression, underscoring its role in shaping the immune tumor microenvironment. This finding is particularly relevant for emerging immunotherapies, as RFC3-associated immune status may impact therapeutic efficacy. Finally, drug sensitivity analysis revealed significant differences between RFC3 expression groups, highlighting RFC3′s potential as a biomarker for predicting treatment response and guiding personalized therapy strategies.

Our study has some limitations that have to be acknowledged. Although this study provides compelling evidence for the role of RFC3 in DLBCL, certain limitations should be acknowledged. The retrospective analysis based on publicly available GEO datasets may introduce selection bias and limit the broader applicability of the findings. This study did not explore the potential regulatory role of microRNAs in modulating RFC3 expression or function in DLBCL. Future investigations incorporating microRNA analyses could provide deeper insights into the post-transcriptional regulation of RFC3 and its impact on lymphoma progression. Additionally, direct functional validation of RFC3′s involvement in tumor progression and immune regulation remains limited, with few experimental studies

addressing these mechanisms. Further investigation using both *in vitro* and *in vivo* models is essential to elucidate the molecular pathways through which RFC3 influences tumor biology and the tumor immune microenvironment. Evaluating RFC3-targeted therapeutic approaches in preclinical settings could also offer valuable translational insights. The small sample size in our study also limits the generalizability of our findings. Further prospective longitudinal studies with larger cohorts are needed to confirm RFC3′s utility as a prognostic biomarker and therapeutic target in clinical practice. Finally, data on important prognostic markers such as KI-67, MYC rearrangement, double-hit, and double-expressor status were either unavailable or insufficient. This limits our ability to assess the association between RFC3 expression and high-risk subtypes fully. Future studies with comprehensive molecular data are needed.

## CONCLUSION

RFC3 is involved in key oncogenic pathways and contributes to an immunosuppressive tumor microenvironment, underscoring its dual potential as both a diagnostic marker and a therapeutic target in various tumours. Overall, our findings indicate that elevated RFC3 expression is a negative prognostic indicator in DLBCL. However, further studies are warranted to validate our findings and improve the prognosis and treatment strategies for DLBCL and other malignancies.

**Abbreviations**

| | |
|---|---|
| **DLBCL** | Diffuse large B-cell lymphoma |
| **RFC3** | Replication Factor C 3 |
| **IPI** | International Prognostic Index |
| **GSEA** | Gene Set Enrichment Analysis |
| **OS** | Overall Survival |
| **PFS** | Progression-free Survival |
| **DSS** | Disease-specific Survival |
| **DFI** | Disease-free Interval |
| **PFI** | Progression-free interval |
| **RFC** | Replication Factor C |
| **GEO** | Gene Expression Omnibus Database |
| **ROC** | Receiver Operating Characteristic |
| **TCGA** | The Cancer Genome Atlas Program |
| **TMB** | Tumor Mutational Burden |
| **MSI** | Microsatellite Instability |
| **ACC** | Adrenocortical Carcinoma |
| **BLCA** | Bladder Urothelial Carcinoma |
| **BRCA** | Breast Invasive Carcinoma |
| **CESC** | Cervical Squamous Cell Carcinoma |
| **CHOL** | Cholangiocarcinoma |
| **COAD** | Colon Adenocarcinoma |
| **ESCA** | Esophageal Carcinoma |
| **GBM** | Glioblastoma Multiforme |

| | |
|---|---|
| **HNSC** | Head And Neck Squamous Cell Carcinoma |
| **LGG** | Lower Grade Glioma |
| **LIHC** | Liver Hepatocellular Carcinoma |
| **LUAD** | Lung Adenocarcinoma |
| **LUSC** | Lung Squamous Cell Carcinoma |
| **OV** | Ovarian Serous Cystadenocarcinoma |
| **PAAD** | Pancreatic Adenocarcinoma |
| **PRAD** | Prostate Adenocarcinoma |
| **READ** | Rectum Adenocarcinoma |
| **SARC** | Sarcom |
| **SKCM** | Skin Cutaneous Melanoma |
| **STAD** | Stomach Adenocarcinoma |
| **TGCT** | Testicular Germ Cell Tumors |
| **UCEC** | Uterine Corpus Endometrial Carcinoma |
| **UCS** | Uterine Carcinosarcoma |
| **KICH** | Idney Chromophobe |
| **LAML** | Acute Myeloid Leukemia |
| **THCA** | Thyroid Carcinoma |
| **THYM** | Thymoma |
| **NPC** | Nasopharyngeal Carcinoma |
| **HCC** | Hepatocellular Carcinoma |

### Funding

The authors received no funding for this work.

### Competing Interests

The authors declare there are no competing interests.

### Author Contributions

- Zuguo Tian conceived and designed the experiments, performed the experiments, analyzed the data, prepared figures and/or tables, authored or reviewed drafts of the article, and approved the final draft.
- Shuiyu Liu performed the experiments, analyzed the data, prepared figures and/or tables, and approved the final draft.
- Chunlan Weng conceived and designed the experiments, analyzed the data, prepared figures and/or tables, authored or reviewed drafts of the article, and approved the final draft.

### Human Ethics

The following information was supplied relating to ethical approvals (*i.e.*, approving body and any reference numbers):

Ethics Committee of Affiliated Hospital of Zunyi Medical University.

## Data Availability

The raw data is available in the Supplemental Files.

## Supplemental Information

Supplemental information for this article can be found online at http://dx.doi.org/10.7717/peerj.20001#supplemental-information.

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
