# Peer review of "Prognostic significance and pathogenesis of RFC3 gene expression in diffuse large B-cell lymphoma"

_PeerJ, doi:10.7717/peerj.20001_

## Round 0.1 · original submission · Major Revisions

The analyses should include MYC rearrangement status. The limitations of small sample size in IHC analysis should be discussed. All of the issues raised by the reviewers should be addressed in detail.

·

Basic reporting

This is innovative work. The data are very interesting. The only limitation is the small number of samples on which immunohistochemical validation was done

Experimental design

Well done

Validity of the findings

Validation should in the near future be done on a larger number of samples

Additional comments

No other comments

Reviewer 2 ·

Basic reporting

This study presents the RFC3 expression gene in DLBCL (Diffuse large B-cell lymphoma). The presentation quality is very low, even English grammar is weak - it contains typos, missed punctuation. The language presentation also should be corrected.
The terms are given without comments, the abbreviations are not shown. The role of microRNA in DLBCL is not discussed. There are multiple parameters, and the database names are given in a mixture.
The literature references are relevant to the study, but do not compete.
The databases and methods should be properly cited in the text.
The figures contain correlations of the single gene expression with multiple poor prognosis parameters.
There are some redundancies in the figures.

Experimental design

This work is based on open data reanalysis. The own experimental part is limited. Ethical letter is not shown - providing explanation instead that the local committee allowed do so.
The study barely meets technical standards.
The methods described have sufficient detail to be replicated. Though the description is very formal, it just mentions database parameters.

Validity of the findings

The findings have limited value. RFC3 gene expression could be used as a prognosis marker. Potentially. Limitations of the study are shown. The discussion section is acceptable.

Additional comments

Overall, this bioinformatics study has limited value.
It needs major revision, providing further details and rephrasing.
The title should have proper upper and lower case presentation of the terms –
“rfc3 expression in dlbcl” - write ‘RFC3’ and ‘DLBCL’ in capital letters, avoid abbreviation, write in full ‘Diffuse large B-cell lymphoma’
Add the word ‘gene’ for RFC3.
Write the term ‘gene expression’, not just ‘expression’.
Avoid passive voice in English – (see, “was investigated…”, “was validated…”, “was assessed…” – write it in direct English – ‘We investigated…’, ‘we validated…’, and so on).
Line 20:’ specifically through’ – rephrase, remove wording ‘specifically’
Line 22: ‘and the analysis result was validated by immunohistochemistry’ – this is not a clear experimental part of the work. At least write it in a separate sentence.
Line 29: ‘adverse clinical characteristics’ – mention these characteristics – the phrase is too common.
Line 33: ‘..in multiple tumor types’ – need to note these tumors. How are they related to DLBCL?
Line 40: ‘(DLBCL( - extra parenthesis. The reference is not to a recent paper by year 2016.
Line 42: ‘standard RCHOP regimen’ – give RCHOP abbreviation in full, comment on it.
Line 45: ‘Replication factor C (RFC) is widely distributed’ – add word ‘gene’, add details about evolutionary conservation.
Line 47: “…Shimada et al. 1999) (Kim & Bril…” – cite fewer papers together, use proper punctuation in the citation.
Line 59: ‘Chae(Chae et al. 2015) reported…’ – write as ‘Chae and colleagues (Chae et al. 2015)’, not using single name.
Line 67: “received R-base chemotherapy” – comment on what the therapy is.
Line 73: ‘R software (version 4.2)’ – need to add reference to R package (at least online).
Line 78: “integrates The data of The…” – remove capital letter, rephrase.
Line 81: - section about Immunohistochemistry - does not need to use the abbreviation IHC in the section title.
Format the section. Data about approval by the Ethics Committee should be reformulated.
Remove the phrase about ‘…waived written informed’ to a supplement or acknowledgement. It is technical information.
Overall, Immunohistochemistry gives no new information to this study.
Line 104: ‘ (IPI)’ - typo
Line 105: R package "ggstatsplot" – need to add reference
Line 114: ‘packages survival and survminer’ – need to add references to these packages.
Line 120: ‘may be involved in the carcinogenesis of DLBCL’ – common phrase, redundant. Remove it.
Line 124: ‘DLBCL, We ‘ – typo
Line 125: ‘IOBR package’ – give the IOBR abbreviation in full, comment on how this package works
Line 127: ‘GSVA R’ - give the abbreviation in full, add comment
Line 129: ‘TISIDB’ – abbreviation in full.
Line 131: ‘Download immune-related genes from…’ – here is a list of parameters, and the sentence is grammatically correct. Rephrase. Avoid new abbreviations like (IIG), (ISG) if used only once in the text.
Line 138: - Drug-susceptibility analysis is not clear – how was it done? By comparison of what and what?
Line 141: ‘GDSC1, GDSC2, and CTRP2’ – these components should be given in full, commented.
Line 145: ‘to provide new ideas for the treatment of DLBCL' – this concluding phrase is too common. Rewrite it separately.
Line 150: ‘(TCGA) and GTEx’ – the databases should be cited. Show GTEx abbreviation in full.
Lines 184-185: all the abbreviated parameters should be explained, given in full – “LDH…, Ann Arbor 185 stages III-IV… ECOG-PS …high IPI”
Line 195 and below: “enriched in several signaling pathways…” – the pathway names are given in capital without comments (E2F TARGETS…. G2M CHECKPOINT…). Add details, explanation about these pathways – are they common for cell growth, related to cancer? Why should this elevation be important? Give pathway names as parameters with proper formatting if it was used formally.
Line 205: “StromalScore, ImmuneScore, and ESTIMATEScore” – describe these parameters – how they appeared, just given in the database? What does it mean?
Next lines 213, 215-217, and below – too long description – list of gene names without any comments. Add spaces, and make an additional table if necessary. It is not good to give several rows of gene names without any readable text.
Line 227-228 – “Drug sensitivity analysis” – this section is not clearly written. How was it done? This paragraph just refers to the figures. Only in the figure legend can one see that the ‘oncopredict R package’ was used. Are all these results based on calculated IC50 values? Assume it should be shown in detail in the methods section, and then interpreted in the Results section.
Line 257: - the references are old (years 2012-2019). Assume the need to use novel data in the discussion.
Line 270: ‘OS and PFS’ – add abbreviations in full.
Lines 272 and then 275 – repeated text pattern “60 years or older at the time of diagnosis, advanced…” – need to be rewritten.
Line 301: ‘n CRC, HCC, and NSCLC’ – give all the abbreviations in full.
Line 319: ‘Tumorpurity’ – use standard writing, distinguishing parameter names and common words.
The Conclusions section has common wording: “Future research should aim to validate these findings…” Write it shorter, use direct language, and proper formatting.

Reviewer 3 ·

Basic reporting

In this study, the authors review the prognostic significance of RFC3 expression in DLBCL and find that RFC3 appears to be significantly overexpressed in DLBCL, which is associated with poor prognosis, adverse clinical characteristics, and tumor microenvironment properties. They suggest that this may potentially serve as a novel prognostic marker and therapeutic target.

Minor Comments:
1. There are multiple spacing issues throughout the manuscript, with spaces omitted between words. Please review and correct these space omissions.

Experimental design

This is a generally well-designed study and a well-written manuscript.

Validity of the findings

This is a generally well-designed study and a well-written manuscript. The results seem valid, and the conclusions are logical, but there are some confounding factors and additional information that likely need to be addressed.

Major Comments:
1. Another major adverse prognostic factor that needs to be included in the analysis is MYC rearrangement status. How many of the 18 DLBCL cases were MYC rearranged, or double or triple hit cases?
2. Also, how many cases were double expressors would also be useful to mention.
3. If there were cases with any of the aforementioned abnormalities, it would be useful to compare RFC3 expression levels to the DLBCL cases without these poor prognostic aberrations, a similar idea to the comparison of GC and non-GC
4. In addition, was there any difference in RFC3 expression between the GC and non-GC types (to each other, not just compared to the control groups)
5. The Ki-67 proliferation index, another pathologic marker for aggressive behavior, may also be helpful to describe the cohort.

Additional comments

Minor Comments:
1. (see basic reporting above)
2. Figure 1C is missing the last L in DLBCL
3. The text for Figure 3 is out of order as compared to the actual figure, making it a little more difficult to follow. Consider reordering the text or the figure so that Figures 3A-3G appear in order in the text.
4. In line 260, the authors mention that the current research corroborates previous findings. Please elaborate or consider altering the phrasing. i.e, do previous findings refer to RFC3 in other tumors or other studies that evaluated it in DLBCL?
5. Figures 1 G-L do not appear to be referenced in the text.
6. The text for the axes of the figures is a little small and difficult to read. Considering changing the font and size to make it larger and clearer.

---

## Round 0.2 · accepted · Accept

The authors have made the necessary revisions or clarifications to satisfy the reviewer’s feedback. Therefore, the manuscript is now deemed suitable for publication.

Reviewer 2 ·

Basic reporting

The paper presents bioinformatics study of gene expression in Diffuse large B-cell lymphoma. Role of RFC3 gene is discussed. The materials are clearly presented meeting journal publication standards.
The materials has been revised. I see detailed answers to my previous reviewing remarks.

Experimental design

This is original bioinformatics research based on gene expression data from public datasets. The methods were described with sufficient details. No more comments to improve.

Validity of the findings

This work present novel finding on RFC3 gene role. It fits the publication standards.

Additional comments

Since it is revised version I have to note that all the reviewing remarks were considered. I have no more critical comments.
Not need additional reviewing round.

Some suggestions -
use 'RFC3 gene' abbreviation in the keywords list too, not just 'Replication Factor C 3'.
Line 42: use lower case letter for 'Pan-cancer'
Lines 121-122: 'Ethics No:KLL-2024-020' - add wording 'Ethics Committee paper' or 'Ethics approve' to the word 'Ethics'.
Line 136: 'Kassambara A et al. 2021' - not need authors' initial 'A'
Lines 187 and 521: '(li. 2024)' - write the reference properly (capital letter, full publication data)
Overall, these comments are on the authors' discretion.